# Secreted Glycosyltransferase RsIA_GT of *Rhizoctonia solani* AG-1 IA Inhibits Defense Responses in *Nicotiana benthamiana*

**DOI:** 10.3390/pathogens11091026

**Published:** 2022-09-09

**Authors:** Danhua Zhang, Zhaoyilin Wang, Naoki Yamamoto, Mingyue Wang, Xiaoqun Yi, Ping Li, Runmao Lin, Zohreh Nasimi, Kazunori Okada, Keiichi Mochida, Yoshiteru Noutoshi, Aiping Zheng

**Affiliations:** 1College of Agronomy, Sichuan Agricultural University, Chengdu 611130, China; 2Rice Research Institute, Sichuan Agricultural University, Chengdu 611130, China; 3Institute of Vegetables and Flowers, Chinese Academy of Agricultural Sciences, Beijing 100081, China; 4Agro-Biotechnology Research Center, The University of Tokyo, Bunkyo-ku, Tokyo 113-8657, Japan; 5Bioproductivity Informatics Research Team, RIKEN Center for Sustainable Resource Science, Yokohama 2300045, Japan; 6Microalgae Production Control Technology Laboratory, RIKEN Baton Zone Program, RIKEN Cluster for Science, Technology and Innovation Hub, Yokohama 2300045, Japan; 7Kihara Institute for Biological Research, Yokohama City University, Yokohama 2440813, Japan; 8School of Information and Data Sciences, Nagasaki University, Nagasaki 852-8521, Japan; 9Graduate School of Environmental and Life Science, Okayama University, Okayama 700-8530, Japan; 10State Key Laboratory of Crop Gene Exploration and Utilization in Southwest China, Chengdu 611130, China

**Keywords:** phytopathogenic fungi, pathogenicity, glycosyltransferases, plant innate immunity, virulence

## Abstract

Anastomosis group AG-1 IA of *Rhizoctonia solani* Khün has a wide host range and threatens crop production. Various glycosyltransferases secreted by phytopathogenic fungi play an essential role in pathogenicity. Previously, we identified a glycosyltransferase RsIA_GT (AG11A_09161) as a secreted protein-encoding gene of *R. solani* AG-1 IA, whose expression levels increased during infection in rice. In this study, we further characterized the virulence function of RsIA_GT. It is conserved not only in Basidiomycota, including multiple anastomosis groups of *R. solani*, but also in other primary fungal taxonomic categories. RsIA_GT possesses a signal peptide (SP) for protein secretion, and its functionality was proven using yeast and *Nicotiana benthamiana*. The SP-truncated form of RsIA_GT (RsIA_GT(ΔS)) expressed in *Escherichia coli-*induced lesion-like phenotype in rice leaves when applied to punched leaves. However, Agrobacterium-mediated transient expressions of both the full-length RsIA_GT and RsIA_GT(ΔS) did not induce cell death in *N. benthamiana* leaves. Instead, only RsIA_GT(ΔS) suppressed the cell death induced by two reference cell death factors BAX and INF1 in *N.*
*benthamiana*. RsIA_GT(ΔS)^R154A D168A D170A^, a mutant RsIA_GT(ΔS) for the glycosyltransferase catalytic domain, still suppressed the BAX- or INF1-induced cell death, suggesting that the cell death suppression activity of RsIA_GT(ΔS) would be independent from its enzymatic activity. RsIA_GT(ΔS) also suppressed the H_2_O_2_ production and callose deposition and showed an effect on the induction of defense genes associated with the expression of BAX and INF1. The transient expression of RsIA_GT(ΔS) in *N. benthamiana* enhanced the lesion area caused by *R. solani* AG-1 IA. The secreted glycosyltransferase, RsIA_GT, of *R. solani* AG-1 IA is likely to have a dual role in virulence inside and outside of host cells.

## 1. Introduction

*Rhizoctonia solani* Khün, the basidiomycete soil-borne fungal pathogen, infects many plants belonging to over 180 genera from 32 families [1,2]. *R. solani* contains 13 anastomosis groups (AG-1 to AG-13), and these anastomosis groups show high genetic diversity [3,4]. This high genetic diversity gives it a broad host range [3]. *R. solani* AG-1 IA is the causal agent of rice sheath blight disease [5,6]. This pathogen rarely produces asexual spores and infects the host through sclerotia or runner hyphae [7,8]. The lesion phenotype in leaf sheath and leaf blades of rice plants infected by this pathogen becomes visible mainly from the late tillering stage to the stem elongation stage. For rice, the global yield losses due to sheath blight typically range from 10–30% to more than 50% when the outbreak is severe [9]. Sheath blight management still requires studying the pathogenesis of *R. solani* AG-1 IA in great detail given the significant loss it causes.

Toxin A, toxin B, and the lethal toxin (LT) of *Clostridium difficile*, hemorrhagic toxin from *Clostridium*
*sordellii*, and the α-Toxin of *Clostridium novyi* are the members of the family of large clostridial cytotoxins [10,11]. Toxin A and B could lead to antibiotic-associated diarrhea and pseudomembranous colitis [12,13,14], and hemorrhagic toxins from *C. sordellii* are related to the gas gangrene syndrome [15,16]. These proteins show glycosyltransferase (GT) activity and could glycosylate GTPases, members of the Rho family, leading to its biologically inactive [11,17,18,19]. Several glycosyltransferase families have been identified and exhibit substantial structural diversity among the families [20]. However, all these glycosyltransferases contain a small, conserved D (aspartic acid) XD motif, which is essential for enzyme activity [20]. Other amino acid residues nearby the DXD motif are conserved among the large clostridial cytotoxins and are important for enzyme activity [21]. Substitution of the DXD motif aspartic acids decreased glucosyltransferase activity by approximately 5000-fold and completely blocked glucohydrolase activity [21]. The double mutant of these aspartic acids also lost its enzyme activity [21]. Substituting Asp 270 and Arg 273 caused glucosyltransferase activity to reduce by about 200-fold and blocked glucohydrolase activity [21]. Studies on important amino acid sites of the large clostridial cytotoxin could promote exploration of the pathogenetic mechanism of GTs in plant pathogens. Glycosylation of effectors and host target proteins by GTs are post-translational modifications and directly affect pathogen virulence and host defense ability [22]. The involvement of glycosylation pathways in pathogenesis in several fungal pathogens has been identified. The glycosyltransferase NleB from *Citrobacter rodentium*, mannosyltransferase Pmt4 from *Ustilago maydis* and ALG3 from *Magnaporthe oryzae*, and the N-glycosylation of the protein disulfide isomerase Pdi1 from *U. maydis*, respectively, affect pathogens’ virulence in varying degrees [23,24,25,26]. Although the effect of GTs on fungal virulence has been extensively explored in many pathogens, there is no report on that in *R. solani* AG-1 IA.

Plant pathogens deliver virulence factors inside and outside of plant cells using various signal motifs and pathways to promote effective colonization [27,28,29]. Gram-negative bacterial pathogens mainly employ type ΙΙ secretion system (T2SS) and type III secretion system (T3SS) to inject virulence factors into host cells [30,31,32]. The components of T2SS are encoded by 12 to 16 genes, which usually appear in a gene cluster [33,34]. The T3SS consists of approximately 25 different proteins, and nine core proteins are conserved in a wide variety of gram-negative plant pathogens [31,35,36]. The T3SS is called the Hrp (hypersensitive response and pathogenicity) system in plant pathogens because the *hrp* genes largely encode components of T3SS, and mutation of any one of the *hrp* genes will lead to the complete loss of the hypersensitive response and their pathogenic ability [37,38,39,40]. There is evidence that plant pathogens utilize the Hrp system directly to transfer Avr effector proteins into plant cells [38]. Rust fungi use haustoria to form a close association with host plant cells, and the haustoria are essential in the delivery of virulence and avirulence factors [41]. The AvrL567 proteins from *Melampsora lini* were the first identified rust Avr proteins delivered directly into the plant cells [42]. AvrL567 proteins contain SPs and are expressed in haustoria and could induce a hypersensitive response (HR) depending on the NBS-LRR L5, L6, or L7 resistance proteins of flax [42]. Some obligate biotrophs, for example, the oomycete downy mildews and the ascomycete powdery mildew pathogens, use the same strategies as the rust fungi to deliver virulence and avirulence factors into host plants [43]. Other hemibiotrophic oomycetes and fungi, such as *Phytophthora*, also utilize haustoria to transport pathogenesis factors during the early stage of infection [44]. Additionally, plant pathogenic *Phytophthora* species and oomycetes produce several effectors possessing N-terminal RXLR-dEER-motif, LxLFAK, ChxC, or the Crinkler motif (CRN) [45]. These motifs are required for these effectors targeting host cells [45]. The haustoria-producing pathogenic fungi, including the barley powdery mildew, the wheat stem rust, and the wheat leaf rust fungi, encode a small group of effectors with signal peptides (SPs) for protein secretion and the conserved Y/F/WxC-motif to promote the transport of these effectors into plant cells [46]. These analyses suggest that secretion signals and systems are crucial for the pathogenicity of plant pathogens.

In our previous study, a draft genome sequence of *R. solani* AG-1 IA was analyzed to predict 965 potential secreted proteins [47]. Among them, 45 secreted protein-encoding genes, which showed two-fold or more significant up-regulation in the early infection stage of *R. solani* AG-1 IA in rice, were tested for potential secreted effectors [47], resulting in the identification of a glycosyltransferase (AG1IA_09161 named RsIA_GT) causing cell death phenotype on rice leaves. Here, we further characterized the virulence function of RsIA_GT using *Nicotiana benthamiana*. Transient expression of RsIA_GT(ΔS), an SP-truncated form of RsIA_GT, or the mutant RsIA_GT(ΔS)^R154A D168A D170A^ could suppress cell death induced by BAX or INF1 in *N*. *benthamiana,* indicating the independency of the suppression effects from the glycosyltransferase activity. The RsIA_GT(ΔS) could also suppress the H_2_O_2_ production, callose deposition, and transcriptional responses of defense-related genes induced by *BAX* or *INF1*. Accordingly, RsIA_GT(ΔS)-expressed *N. benthamiana* leaves showed enhanced susceptibility to *R. solani* AG-1 IA.

## 2. Results

### 2.1. RsIA_GT Is Widely Distributed in Late-Diverging Fungi and Highly Conserved among R. solani Isolates

RsIA_GT encodes a 346-aa protein that contains a glycosyltransferase domain (57–241 aa). It belongs to ‘glycosyltransferase family 2’ in CAZymes classification [http://www.cazy.org, accessed on 26 July 2022]. To see the distribution and conserved degree of the homologs of this protein in fungi, BLASTP searches against the non-redundant protein database ‘nr’ were performed with E-value of 1.0 × 10^−10^ and the minimum query coverage of 75%. We found homologous proteins in the fungal categories of Basidiomycota [386 proteins in 293 species], Ascomycota [881 proteins in 642 species], Mucoromycota [78 proteins in 63 species], Zoopagomycota [11 proteins in 10 species], and Chytridiomycota [14 proteins in 8 species]. In contrast, no homologous proteins were found in the earliest-diverging categories Microsporidia and Cryptomycota. These results implicate that fungi acquired and evolved this glycosyltransferase during the process of adaptation and co-evolution between pathogen fungi and hosts. To check the molecular evolutionary relationship of the glycosyltransferase in basidiomycetes, we analyzed selected 16 glycosyltransferase-homologous proteins in 13 fungal species in a phylogenetic tree. The maximum likelihood tree formed two major clusters for (1) proteins in Ceratobasidiaceae and Tulasnellaceae, and (2) proteins in other Agaricomycetes (Figure 1). As expected, *R. solani* anastomosis groups are located in the same clade with closer positions (Figure 1). The RsIA_GT showed high identify and similarity with two proteins in *R. solani* AG-1 IB (CEL61542.1, 93% identity, 95% similarity; CCO37913.1, 88% identity, 91% similarity), one protein in *R. solani* AG-3 (EUC56006.1, 89% identity, 93% similarity), and one protein in *R. solani* AG-8 (KDN33818.1, 88% identity, 92% similarity) (Appendix A). We could find a highly homologous sequence (IC-TC_DN321_c0_g1_i1) with RsIA_GT by a TBLASTN search against the AG-1 IC transcriptome contigs assembled by Yamamoto et al. [48]. It indicates the evolutionary conservation of RsIA_GT orthologues in AG-1.

### 2.2. Signal Peptide (SP) of RsIA_GT Is Functional

The protein RsIA_GT was predicted to contain an SP by SignalP 3.0 (Figure 2a). To evaluate the functionality of this SP, we used the yeast secretion system (YST). The YST comprises the yeast strain YTK12 and the vector pSUC2 carrying the sequence encoding invertase without the native SP. YTK12 can only grow on the YPRAA medium containing raffinose as the sole carbon source after being transformed by a recombinant pSUC2 introduced with the functional SP [49]. The DNA fragment for the SP of RsIA_GT was inserted in the upstream region of the invertase gene as in-frame fusion in pSUC2, and it was then transformed into the YTK12.

The SPs of Mg87 protein in *M. oryzae* and Avr1b in *P. sojae* served as the negative and the positive control, respectively [50,51]. The YTK12 transformed by pSUC2 containing the SP of Avr1b showed growth on the YPRAA medium, indicating that the SP could lead to the secretion of invertase and hydrolyze raffinose to support the growth of YTK12 (Figure 2b). In contrast, the negative control containing the SP of Mg87 and the vector-free strain YTK12 did not grow on the YPRAA medium (Figure 2b). The YTK12 carrying pSUC2 with RsIA_GT SP could grow on YPRAA (Figure 2b). The color reaction was also applied to further check the function of the SP of RsIA_GT. Each of the transformed yeast strains was grown in a liquid YPDA medium and the colorless 2, 3, 5-triphenyltetrazolium chloride (TTC) was added into each culture to test if it can be reduced into the red insoluble 1, 3, 5-triphenylformazan (TPF) by the invertase. As a result, the SP of RsIA_GT and the positive control but not the negative controls presented red (Figure 2b).

INF1 is a well-characterized elicitin with SP produced by *P. infestans*, which can infect potato and tomato plants [52]. When INF1 was transiently expressed in *N. benthamiana* leaf cells by the *Agrobacterium* syringe infiltration method, HR cell death was observed (Figure 2c). The SP is required for this INF1-inducing HR cell death [53,54,55]. Therefore, we substituted the native coding sequence of SP of INF1 with that of RsIA_GT to produce a chimeric protein. The chimeric protein was named SP(RsIA_GT)-C-INF1. The transient expression of SP(RsIA_GT)-C-INF1 could also induce HR in the leaves of *N. benthamiana* (Figure 2c). The RsIA_GT and GFP as negative controls could not induce the HR (Figure 2c). Trypan blue staining confirmed the cell death in the area of the necrotic lesions induced by INF1 and SP(RsIA_GT)-C-INF1 (Figure 2d). Western blot confirmed the expression of these proteins in *N. benthamiana* (Figure 2e). These results proved that the SP of RsIA_GT is functional.

### 2.3. RsIA_GT of R. solani AG1-IA Induces Cell Death and Leaf Chlorosis in Rice

The SP of the amino acid chain is usually cleaved off when the protein is exported through the membrane [56]. The RsIA_GT(ΔS) without SP, which is the tentative mature form of RsIA_GT, contains 320 amino acids and has a molecular weight of 35.2 kDa. The DNA fragment of RsIA_GT(ΔS) was fused with the downstream of the coding sequence of maltose-binding protein (MBP, 42.5 kDa) to produce a chimeric protein RsIA_GT(ΔS)-MBP (about 77.74 kDa). The chimeric protein RsIA_GT(ΔS)-MBP was successfully expressed in *Escherichia coli* BL21 (Figure 3a). To examine whether the protein RsIA_GT(ΔS) has a virulent activity in rice, we put the protein extract prepared from *E. coli* expressing RsIA_GT(ΔS)-MBP protein onto rice leaves pierced with a punch. The protein extracted from *E. coli* expressing MBP was used as a control. As shown in Figure 3b, the RsIA_GT(ΔS)-MBP-containing protein extract treatment induced chlorosis in rice leaves. The lesion and yellowing area was 60 times than that of the control (Figure 3b,c). In the previous study, the full-length RsIA_GT protein prepared with *E. coli* was shown to produce chlorosis in rice [47]. In this study, we confirmed the virulence activity of RsIA_GT in rice again and showed that its activity is in the coding region except for SP.

### 2.4. RsIA_GT(ΔS) Suppresses INF1- and BAX-Induced Cell Death in N. benthamiana

We tested if the RsIA_GT protein can induce necrosis or yellowing in the leaves of *N*. *benthamiana*. The full-length RsIA_GT, RsIA_GT(ΔS), *GFP*, *INF1*, or *BAX* were transiently expressed in the leaves of *N. benthamiana* by the syringe infiltration method using *Agrobacterium*. The results showed that neither RsIA_GT nor RsIA_GT(ΔS) could induce cell death (Figure 4a). Trypan blue did not stain the infiltrated area, confirming the no HR induction by RsIA_GT and RsIA_GT(ΔS) (Figure 4b). The western blot confirmed the expression of these proteins (Figure 4c).

Next, we tested whether RsIA_GT possesses a suppressing activity to plant immunity. For this aim, the RsIA_GT or RsIA_GT(ΔS) was co-infiltrated with INF1 or BAX into *N. benthamiana* leaves to see if cell death induced by INF1 or BAX was suppressed or not because INF1- and BAX-induced cell death in *N. benthamiana* is related to the disease resistance response [57,58]. As a result, RsIA_GT(ΔS) but not RsIA_GT suppressed cell death induced by INF1 and BAX (Figure 4d). Trypan blue staining supported these results (Figure 4e). The western blot confirmed co-expression of RsIA_GT, RsIA_GT(ΔS), and GFP with BAX or INF1 (Figure 4f). The inhibition of BAX- or INF1-induced cell death suggests the immune suppressive activity of RsIA_GT(ΔS) in *N. benthamiana*. Since full-length RsIA_GT did not exhibit inhibitory activity to the HR cell death, translocation of RsIA_GT into cytoplasm space might be required for cell death suppression in *N. benthamiana*.

### 2.5. RsIA_GT(ΔS) Inhibits H_2_O_2_ Production and Callose Deposition and Affects Expression of Defense-Related Genes

As mentioned above, RsIA_GT(ΔS) suppresses HR cell death induced by INF1 and BAX. In order to test whether the productions of H_2_O_2_ and callose triggered by INF1 and BAX are also inhibited by RsIA_GT(ΔS), we performed DAB and aniline blue staining on *N. benthamiana* leaves after co-expression of RsIA_GT(ΔS) with *INF1* or *BAX*. In the leaves expressed with *BAX* or *INF1*, the stained areas with DAB (Figure 5a,b) or aniline blue (Figure 5c,d) were detected 12 h and 24 h after the agroinfiltrations, respectively. When RsIA_GT(ΔS) was co-expressed with *BAX* or *INF1*, these signals were almost undetected (Figure 5a–d). The GFP was used as a negative control; however, no suppression of the signals was observed (Figure 5a–d). These results indicate that RsIA_GT(ΔS) suppressed the H_2_O_2_ production and callose deposition induced by *BAX* and *INF1*.

Furthermore, a quantitative reverse transcriptase-polymerase chain reaction (qRT-PCR) was performed to monitor the expressions of the defense-related genes after the transient expression of RsIA_GT(ΔS) with *BAX* or *INF1* in *N. benthamiana* leaves. We selected *respiratory burst oxidase homolog B* (*RbohB), pathogenesis-related gene 4a* (*PR4a*), and *WRKY12* as marker genes of defense response; *RbohB* is involved in the production of H_2_O_2_ [59,60], *WRKY12* is a transcription factor that activates defense-related genes [61,62], and *PR4a* gene is activated by jasmonic acid-dependent defense response pathway [63,64]. The leaf samples were collected at 12 and 36 h after co-infiltration of *Agrobacterium* harboring RsIA_GT(ΔS) and *BAX* or *INF1* and subjected to the qRT-PCR analysis. The transcript abundances of *PR4a* and *WRKY12* at 12 h and *RbohB* at 36 h in the leaves where RsIA_GT(ΔS) and *BAX* or *INF1* were co-expressed were significantly reduced compared with the control with *GFP* instead of RsIA_GT(ΔS) (Figure 5e,f). In the BAX-induced defense responses, the reduction rate of *RbohB* and *WRKY12* transcript levels was over ten and five times to those of the control, respectively (Figure 5e). The *PR4a* expression level reduced over ten times compared to the control (Figure 5e). As for *INF1*, *RbohB*, *WRKY12*, and *PR4a,* the reduction rates were over three times, nearly two times, and over two times, respectively, compared to the control (Figure 5f). These results indicate that RsIA_GT(ΔS) interferes with transcriptional responses of defense genes during HR.

### 2.6. The Suppression Activity of RsIA_GT(ΔS) for the BAX- or INF1-Induced HR Cell Death Is Possibly Independent from Its Glycosyltransferase Activity

Toxins of the large clostridial cytotoxins have glycosyltransferase activity [21]. They take a glucosyl moiety from UDP-glucose and attach it to the target proteins, such as Rho and Ras GTPases. The DXD motif and the close vicinity amino acid residues of the DXD motif in these toxins are conserved and essential for the glycosyltransferase activity [11,12,13,14,15]. To check whether RsIA_GT(ΔS) contains these conserved residues and the DXD motif, multiple alignment of the amino acid sequence of RsIA_GT(ΔS) with the large clostridial cytotoxins was performed. As shown in Figure 6a, RsIA_GT(ΔS) contains the DXD motif [D168, L169, and D170] and the arginine residue [R154], which are conserved in these toxins.

To examine if the potential glycosyltransferase activity of RsIA_GT(ΔS) is required for the suppression of immune responses induced by *BAX* or *INF1*, we prepared an agroinfiltration construct for a mutant version of RsIA_GT(ΔS), of which conserved three amino acid residues (R154, D168, and D170) were substituted with alanine. The leaves were collected at four days after the agroinfiltration, and necrotic lesions (Figure 6b) and HR cell death confirmed by trypan blue staining (Figure 6c) were detected. Similar to RsIA_GT(ΔS), RsIA_GT(ΔS)^R154A^
^D168A^
^D170A^ also suppressed the necrosis induced by both *BAX* and *INF1* (Figure 6b). GFP used as a negative did not suppress cell death. Trypan blue staining further confirmed the results (Figure 6c). Western blot analysis confirmed the co-expression of RsIA_GT(ΔS), RsIA_GT(ΔS)^R154A^
^D168A^
^D170A^, and GFP with BAX or INF1 (Figure 6d). These results suggested that the suppression effect of RsIA_GT(ΔS) on HR cell death induced by *BAX* or *INF*1 is possibly independent of its potent glycosyltransferase activity.

### 2.7. Transient Expression of RsIA_GT(ΔS) Enhances the Susceptibility of N. benthamiana to R. solani AG-1 IA

To examine whether the overexpression of RsIA_GT(ΔS) in host plants enhances the susceptibility to *R. solani* AG-1 IA, we transiently expressed the RsIA_GT(ΔS) protein in the leaves of *N*. *benthamiana* by the agroinfiltration method. After two days, the leaves were inoculated with a 2 mm^2^ active mycelial plug of *R. solani* AG-1 IA. GFP was used as a negative control. In the leaves infiltrated with Agrobacterium harboring RsIA_GT(ΔS), significant necrotic lesions were observed compared with those of the *GFP* control (Figure 7a,b). Quantitative measurement of the lesion area demonstrated that the lesions observed in the leaves with transient expression of RsIA_GT(ΔS) were over 27 times than that of the control (Figure 7c). This result showed that the disease symptom caused by *R. solani* AG-1 IA was promoted in the leaves expressing RsIA_GT(ΔS). The consistency of it with the interference in the plant defense responses indicates the virulence function of RsIA_GT(ΔS).

## 3. Discussion

Glycosyltransferases and mannosyltransferases play different roles from each other during host infection by pathogens. Several glycosyltransferases and mannosyltransferases function as virulence factors that have been identified from many fungal pathogens [24,26,65,66]. However, similar virulence factors have not yet been reported in *R.solani* AG-1 IA. Our previous and current study showed that a secreted protein, RsIA_GT, having a glycosyltransferase domain, induced chlorosis in rice leaves. This result clearly demonstrated the virulence activity of this secreted protein. Although, it seems not to meet specific criteria as an effector protein predicted by several algorithms [66], RsIA_GT was proven to have a virulent function during infection of *R. solani* AG-1 IA. Since *R. solani* AG-1 IA is a necrotrophic fungal pathogen, induction of chlorosis or necrosis would facilitate its infection.

The *N*-glycosylation of the apoplastic effector PsXEG1 from *P. sojae* could protect it against the degradation by the host apoplastic aspartic protease GmAP5 [67]. In *U. maydis*, the *N*-glycosylation of the protein disulfide isomerase Pdi1 is required for the secretion of the pathogens’ virulence factors [25]. The *O*-mannosyltransferase Pmt4 of *U. maydis* and promoting appressorium formation by *O*-mannosylation Msb2 could also promote the fungal spreading inside of infected leaves by *O*-mannosylated Pit1 [24]. In *M. oryzae*, ALG3-mediated *N*-glycosylation of effector Slp1 is critical for its activity during host infection [26]. RsIA_GT is likely to glycosylate effector proteins before and after being secreted from *R. solani* AG-I IA to promote the infection to host plants. The KOBITO1 is a glycosyltransferase-like protein and could regulate the plasmodesmatal permeability and stomatal patterning in *Arabidpsis thaliana* [68]. Finally, we speculate that the enhanced pathogenicity of *R. solani* AG-1 IA by RsIA_GT(ΔS) might be caused via the regulation of plasmodesmatal permeability and stomatal patterning.

We also found that RsIA_GT could suppress H_2_O_2_ production and callose deposition and affect the expression of genes related to defense responses induced by *INF1* or *BAX* when the SP-truncated RsIA_GT was expressed in *N. benthamiana* leaves. The transient expression of RsIA_GT(ΔS) could also promote the symptom formation caused by *R. solani* AG-1 IA in *N. benthamiana*. Surprisingly, the native form of RsIA_GT showed no such immune-suppressing activity. RsIA_GT in plant apoplastic space induces necrosis; however, RsIA_GT inside plant cells can suppress defense responses. Suppression of host immunity by effector proteins or bioactive chemical molecules is required for the infection of biotrophic pathogens. The potential biotrophic interaction of *R. solani* at the early infection stage was recently demonstrated in the interaction with *Brachypodium distachyon*. Thus, the finding of immune-suppression activity of particular *R. solani*-secreted proteins is reasonable [69,70]. However, further study is needed to verify if such biotrophic stage also exist in *N. benthamiana* because pretreatment of salicylic acid did not attenuate virulence of *R. solani* in *A**. thaliana* [71].

Our results suggest that RsIA_GT should work inside host cells at least in *N. benthamiana*. *R. solani* AG-1 IA may directly release RsIA_GT into host cells; otherwise. the secreted RsIA_GT from *R. solani* into apoplast may be incorporated into host cells possibly by hijacking the host protein sorting mechanism. Further analysis is needed to directly observe the presence of RsIA_GT protein in host cells during infection of *R. solani* AG-1 IA.

The phylogenetic analysis showed that homologs of RsIA_GT are widely distributed in basidiomycetes fungi and highly conserved across different *R. solani* anastomosis groups. These secreted glycosyltransferases homologous to RsIA_GT may also function as a virulence factor in these plant-associated fungi. Further analysis in the conservation of essential amino acid residues of glycosyltransferase catalytic domain among these homologous proteins and validation as a virulence factor would reveal commonality or difference in infection strategy of these fungi.

HR, including cell death, is vital to suppress the progression of pathogens [72]. The defense response induced by *INF1* and *BAX* was shown to inhibit the spreading of pathogen hyphae in hosts [73,74,75]. The H_2_O_2_ produced during HR is harmful to pathogens [76,77]. The expression of RsIA_GT(ΔS) suppressed the various defense responses triggered by *INF1* and *BAX*. We initially speculated that the glycosylation of the target protein by RsIA_GT(ΔS) is a possible mechanism for its immune-suppressing activity. However, RsIA_GT(ΔS)^R154A D168A D170A^, a mutated form of RsIA_GT(ΔS) that possibly lacks the glycosyltransferase activity, was shown to suppress the HR cell death induced by *BAX* and *INF1*. This result indicates that RsIA_GT(ΔS) suppresses plant immune responses by inhibiting the target protein independent of its enzymatic activity. It has been shown that INF1 could form a complex with SERK3/BAK1 and NbLRK1 to promote cell death and resistance to pathogens in *N. benthamiana* [78,79]. BAX/BAK is necessary for mediating cell death [80,81,82]. Pattern recognition receptor activates Rboh to produce reactive oxygen species, possibly leading to HR cell death and up-regulated expressions of defense-related genes through the activation of mitogen-activated protein kinases. Because RsIA_GT(ΔS)-suppressed H_2_O_2_ production and interference in defense gene expressions leads to callose deposition, it may target the pattern recognition receptor complex. Further analysis will verify this possibility and highlight the role of RsIA_GT(ΔS) in determining fungal pathogenicity and fungus–host interactions.

## 4. Materials and Methods

### 4.1. Strains, Plant Materials, and Culture Conditions

The *R. solani* national standard strain AG1IA was cultured in PDB medium (200 g potato, 20 g sucrose, add distilled water to 1000 mL) at 28 °C for over two nights to inoculate rice leaves for RNA extraction. Competent *E. coli* DH5α was used to construct vectors. The transformed strains were cultured at 37 °C in Luria Bertani (LB) medium. Competent *E. coli* BL21 was used for prokaryotic expression, and these transformed strains were cultured at 37 °C in LB medium. *A. tumefaciens* GV3101 was used for transient expression in *N. benthamiana* and cultured in YEP medium at 28 °C. *N.*
*benthamiana* was cultured at 18-h/6-h night-day photoperiods at 23 °C for transient expression.

### 4.2. Plasmid Construction

The PCR fragments of the coding sequence of RsIA_GT(ΔS) (without SP) were fused to the downstream of the coding sequence of MBP located in the prokaryotic expression vector to produce the chimeric protein RsIA_GT(ΔS)-MBP. The PCR fragments of *INF1*, *BAX*, *GFP*, RsIA_GT(ΔS), and RsIA_GT were inserted into the *p35S-FLAG* vector to produce *35S_pro_::INF1-FLAG*, *35S_pro_::BAX-FLAG*, *35S_pro_::GFP-FLAG*, *35S_pro_::RsIA_GT(**ΔS)-FLAG*, and *35S_pro_::RsIA_GT-FLAG* for cell death inhibition and western blot analysis. The sequence encoding the SP of RsIA_GT was fused with the upstream region of the coding sequence of invertase in the pSUC2 vector and the upstream region of the coding sequence of *INF1* (without the native SP) to investigate the function of the SP. The sequence encoding the key amino acid residues of the catalytic domain of RsIA_GT(ΔS) was replaced with alanine and inserted into the p35S vector to produce the transient expression construct for the mutant RsIA_GT(ΔS)^R154A D168A D170A^. All primers used are shown in Appendix A.

### 4.3. RNA Isolation and qRT-PCR Analysis

The *N. benthamiana* leaves, which transiently expressed target proteins, were sampled at different time points. The collected leaves were ground in liquid nitrogen, and then RNAs were extracted according to the manufacturer’s instruction of EZ-10 Total RNA Mini-Preps Kit (Sangon Biotech, Shanghai, China). The 1st-strand cDNA was synthesized according to the manufacturer’s instructions for HiScript II One Step RT-PCR Kit (Vazyme Biotech, Nanjing, China). The RNA extraction of *R. solani* AG-1 IA was conducted by the abovementioned method. These cDNAs were used for qRT-PCR assays. The relative expression values were determined using reference genes, *actin* in *R. solani* AG-1 IA and *EF1a* in *N. benthamiana,* respectively, and calculated with the formula 2^–^^ΔΔCt^.

### 4.4. Trypan Blue Staining for Detecting Cell Death

Dead cells could be stained with trypan blue, and we used this method to evaluate the cell death-inducing activity of proteins. The leaves of *N. benthamiana* were collected after two days inoculated by *A. tumefaciens* strain GV3101 carrying different vectors. The leaves were immersed in the stationary liquid (acetic acid: ethanol = 3:1) for two days, the stationary liquid was discarded, and the leaves were washed with distilled water twice. Then, the leaves were immersed in the staining solution (0.67 g trypan blue, 100 mL phenol, 100 mL lactic acid, 100 mL glycerol, 100 mL distilled water, and 600 mL ethanol) for one day at room temperature. Then, the staining solution was replaced with a destaining solution (chloral hydrate 2 g/mL) and incubated for two days.

### 4.5. 3,3′-Diaminobenzidine Staining for Detecting H_2_O_2_

Hydrogen peroxide is produced during the plant immune response and could be detected by staining with 3, 3′-diaminobenzidine (DAB), which forms a brown compound. The *N. benthamiana* leaves under transient expression of proteins were collected and immersed in the DAB staining solution (1 mg/mL) for two days at room temperature. Then, the leaves were placed in 95% alcohol and microwaved for 1 min until completely discolored. The leaves were immersed in the destaining solution (chloral hydrate 2 g/mL) until the background color disappeared. We washed the leaves with distilled water twice and imaged with a microscope.

### 4.6. Aniline Blue Staining for Detecting Callose Staining

Callose is also a product produced during the plant immune response and could be observed by aniline blue staining. The *N. benthamiana* leaves transiently expressed proteins were collected and immersed in the destaining solution (80 mL lactic acid, 80 mL glycerine, 80 mL phenol, 80 mL distilled water, and 640 mL ethanol) at 65 °C under a dark place. After decolorization, the leaves were washed with 50% ethanol twice. Then, the leaves were placed into the staining solution (K_2_HPO_4_·3H_2_O 17.1165 g, pH 9.5, aniline blue 0.05 g, distilled water to 500 mL) for 1 h. The leaves were placed on a slide adding 50% glycerol and detected callose by a fluorescence microscope.

### 4.7. Analysis of Cell Death Suppression in N. benthamiana

Strains of *A. tumefaciens* GV3101, respectively, carrying vectors *35S_pro_::RsIA_**GTs-FLAG*, *35S_pro_:**:**RsIA_GT(**ΔS)**-FLAG*, and *35S_pro_::GFP-FLAG* were inoculated in YEP medium containing appropriate antibiotics for 48 h at 28 °C. The bacterial cultures above were, respectively, collected and re-suspended in MMA buffer [10 mM MgCl_2_, 10 mM MES (pH 5.7), 200 μM acetosyringone] to a final OD_600_ of 0.5. All suspensions were incubated for 2–3 h in the dark prior to infiltration. Then, *N. benthamiana* leaves were infiltrated with the MMA incubation buffer containing *35S_pro_::GFP-FLAG*, *35S_pro_::RsIA_GT(**ΔS)*, and *35S_pro_::RsIA_GT-FLAG,* respectively. The infiltration sites were challenged one day after with the MMA incubation buffer containing *A. tumefaciens* carrying *35S_pro_::INF1-FLAG* or *35S_pro_::BAX-FLAG*. *A. tumefaciens* strain GV3101 carrying the vector *35S_pro_::GFP-FLAG* was used as a negative control. Cell death status was observed and compared over the next few days.

### 4.8. Construction of Phylogenetic Tree

BLASTP searches of amino acid sequence of RsIA_GT against the GenBank [83] non-redundant protein database ‘nr’ were performed to find homologous proteins with RsIA_GT. Then, selected 16 homologous protein sequences in 15 basidiomycete strains were retrieved from GenBank and imported into MEGA 7 [84]. A multiple sequence alignment was prepared using MUSCLE [85] and applied for the construction of a phylogenetic tree with the maximum likelihood method using the Whelan and Goldman + Freq. model.

### 4.9. Protein Extraction and Western Blotting

Proteins were extracted, according to Moffett et al. [86] described, and made a little modification. Approximately 0.5 g leaf tissues of *N. benthamiana,* respectively, expressed *RsIA_GT-FLAG*, *RsIA_GT(ΔS)-FLAG*, *INF1-FLAG*, *BAX-FLAG*, and *GFP-FLAG* were harvested at two days post infiltration and powdered in liquid nitrogen using mortars and pestles. Resultant powders were re-suspended by 1.0 mL of extraction buffer [25 mM Tris-HCl (pH 7.5), 1 mM EDTA, 150 mM NaCl, 10% glycerol, 5 mM dithiothreitol, and 2% polyvinylpolypyrrolidone] and centrifuged at 4 °C for 15 min. The supernatant containing the transiently expressed proteins was, respectively, collected into new centrifuge tubes. The extracted proteins were separated by electrophoresis in 12.5%, 10%, or 15% sodium dodecyl sulfate–polyacrylamide gels. Then, the separated proteins were transferred to nitrocellulose membranes at 25 volts for 10 min. The nitrocellulose membranes were washed three times by TBS-T buffer [50 mM Tris-HCl, (pH 7.5), 0.05% Tween 20, 150 mM NaCl] and then blocked with TBS-T buffer containing 5% milk for 1.5 h at room temperature. The membranes were washed three times by TBS-T buffer and then incubated at 4 °C overnight with TBS-T buffer containing 5% milk and an anti-FLAG antibody (1:2000 dilution, Sangon Biotech, Shanghai, China). The membranes were washed three times with TBS-T buffer and then incubated with the mouse-anti-rabbit second antibody for 1 h at room temperature. Then, these membranes were visualized with BeyoECL Star kit (Beyotime Biotechnology, Shanghai, China) and photographed on X-ray films.

## 5. Conclusions

Our study provided evidence that RsIA_GT(ΔS), a secreted glycosyltransferase from *R. solani* AG-1 IA, inhibits plant defense responses and promotes the pathogen infection. RsIA_GT(ΔS)^R154A D168A D170A^, the mutant of glycosyltransferase catalytic domain, still inhibited cell death, H_2_O_2_ production, and callose deposition induced by *INF1* or *BAX* in *N.*
*benthamiana*. Although further studies on the GT activity of the mutant RsIA_GT(ΔS)^R154A D168A D170A^ are needed, this suggests that RsIA_GT(ΔS) might contribute to virulence independently of its enzymatic activity. Future studies on receptors or interacting proteins of RsIA_GT could be helpful to elucidate the pathogenic mechanism of *R. solani* AG-1 IA and provide a valuable target for disease resistance breeding.

## Figures and Tables

**Figure 1 pathogens-11-01026-f001:**
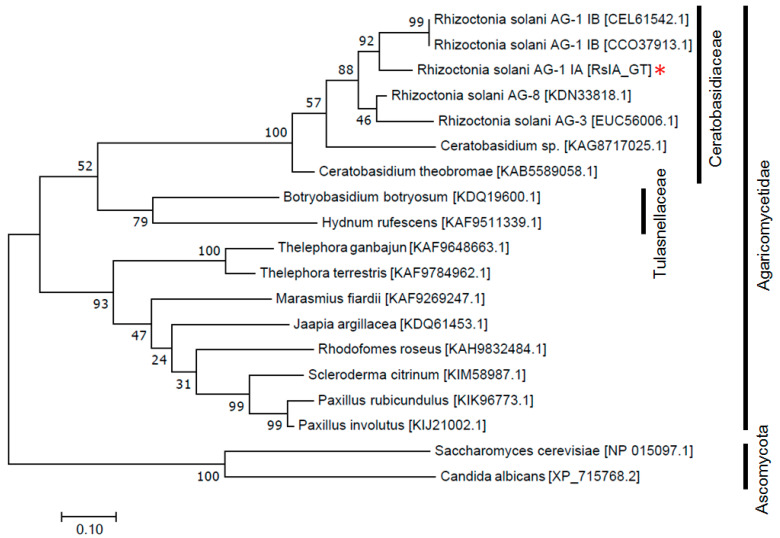
The phylogenetic tree of RsIA_GT and glycosyltransferase in other basidiomycetes. The tree with the highest log likelihood (−7084.07) is shown. Initial tree(s) for the heuristic search was obtained automatically by applying Neighbor-Join and BioNJ algorithms to a matrix of pairwise distances estimated using the Whelan and Goldman + Freq. model and then selecting the topology with superior log likelihood value. The tree is drawn to scale, with branch lengths measured in the number of substitutions per site. Numerals on branches indicate probabilities with 1000 bootstrap trials. A red asterisk indicates RsIA_GT.

**Figure 2 pathogens-11-01026-f002:**
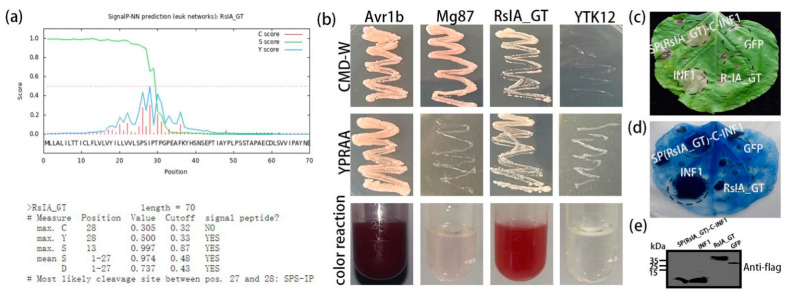
Prediction and functional validation of the signal peptide (SP) of RsIA_GT: (**a**) The score provided by SignalP 3.0 for the SP analysis of RsIA_GT. The 27 amino acid residues of the N-terminus of RsIA_GT were predicted as an SP. (**b**) Functional validation of SP of RsIA_GT by yeast secretion system (YST). (**top**) Yeast strains containing *pSUC2* vectors with or without functional SP could grow on the CMD-W medium containing sucrose and glucose as carbon sources but not tryptophan for vector selection. The SPs of Mg87 protein of *M. oryzae* and Avr1b protein of *P. sojae* are the negative and the positive control, respectively. (**middle**) Only yeast containing *pSUC2* vectors with functional SP could grow on the YPRAA medium. (**bottom**) Validation of invertase activity outside the yeast cells with 2, 3, 5-triphenyltetrazlium chloride (TTC). The supernatants of the liquid culture medium of each yeast strain were supplemented with colorless TCC. The invertase activity can be detected as the red color of 1,3,5-triphenylformazan (TPF) by the conversion of TTC to TPF with invertase enzymatic activity. (**c**,**d**) Functional validation of the SP of the RsIA_GT using INF1-induced cell death in *N. benthamiana* leaves as an indicator by transient expression of *Agrobacterium* syringe infiltration method. The photograph was taken at four days after the infiltration (**c**), and it was subsequently stained with trypan blue (**d**). RsIA_GT without SP and green fluorescent protein (GFP) served as the negative control. INF1 served as the positive control. (**e**) Western blot confirmation expression of INF1, SP(RsIA_GT)-C-INF1, RsIA_GT, and GFP using anti-FLAG antibody.

**Figure 3 pathogens-11-01026-f003:**
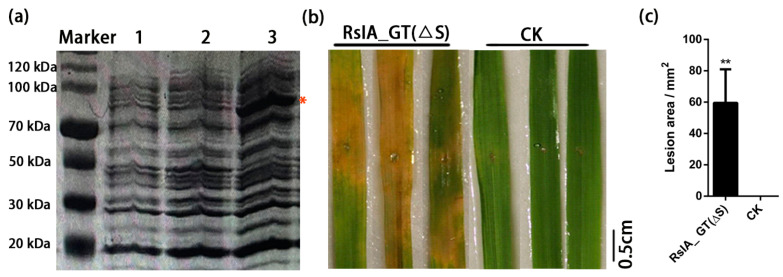
RsIA_GT(ΔS) induces chlorosis after the treatment with wounded rice leaves: (**a**) SDS-PAGE of the soluble fraction of protein extracts of the uninduced (lanes 1 and 2) and the induced (lane 3) *E. coli* BL21 strain expressing RsIA_GT(ΔS)*-MBP*. A red asterisk shows the expected size of the target protein (77.74 kDa). (**b**) Rice leaves treated with the extracted proteins (20 μg/mL, not purified) of *E. coli* expressing plasmids harboring RsIA_GT(ΔS)*-MBP* or *MBP* (CK: the negative control). The photographs were taken at five days after the treatments. (**c**) Lesion-like area in rice leaves induced by treating the protein extract of *E. coli* BL21 expressing RsIA_GT(ΔS)*-MBP* at five days after the treatment. The areas were quantified using ImageJ software. Data represent mean ± SE of three independent biological replicates (** *p* < 0.01, Student’s *t*-test).

**Figure 4 pathogens-11-01026-f004:**
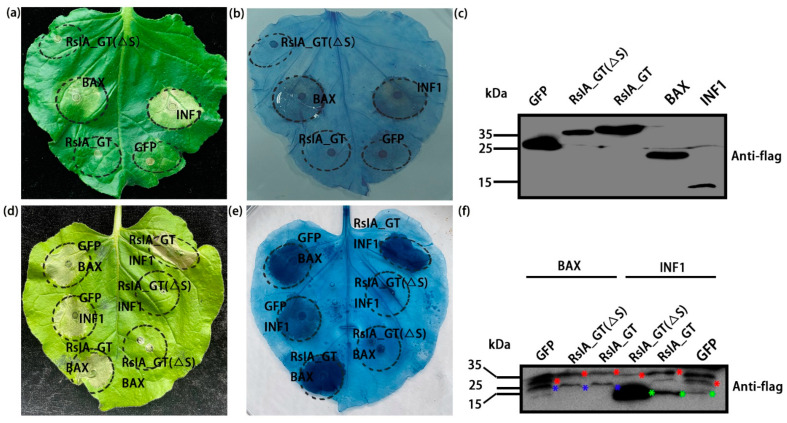
Evaluation of the cell death-inducing activity and the immunity-suppressing activity of RsIA_GT and RsIA_GT(ΔS) in *Nicotiana benthamiana* using *Agrobacterium-*mediated transient expression analysis: (**a**,**b**) Visible necrotic phenotype (**a**) and trypan-blue stained cell death (**b**) induced by the transient expressions of RsIA_GT, RsIA_GT(ΔS), *BAX*, and *INF1*. *GFP* served as the negative control. The photograph was taken at four days after the infiltration. Both RsIA_GT and RsIA_GT(ΔS) did not induce cell death in *N. benthamiana*. (**c**) Western blot confirmation of expression of RsIA_GT, RsIA_GT(ΔS), BAX, INF1, and GFP using anti-FLAG antibody. (**d**,**e**) Visible necrotic phenotype (**d**) and trypan-blue stained cell death (**e**) after the co-expression of RsIA_GT and RsIA_GT(ΔS) with *BAX* or *INF1*. *GFP* served as the negative control. The photograph was taken at four days after the infiltration. (**f**) Western blot confirmation of co-expression of RsIA_GT, RsIA_GT(ΔS), and GFP with BAX or INF1 using anti-FLAG antibody. The blue asterisks indicate the protein bands of BAX. The green asterisks indicate the protein bands of INF1. The red asterisks indicate the protein bands of the correct size. Both RsIA_GT and RsIA_GT(ΔS) did not induce cell death in *N. benthamiana*. Only RsIA_GT(ΔS) suppressed the HR cell death induced by BAX and INF1 in *N. benthamiana*.

**Figure 5 pathogens-11-01026-f005:**
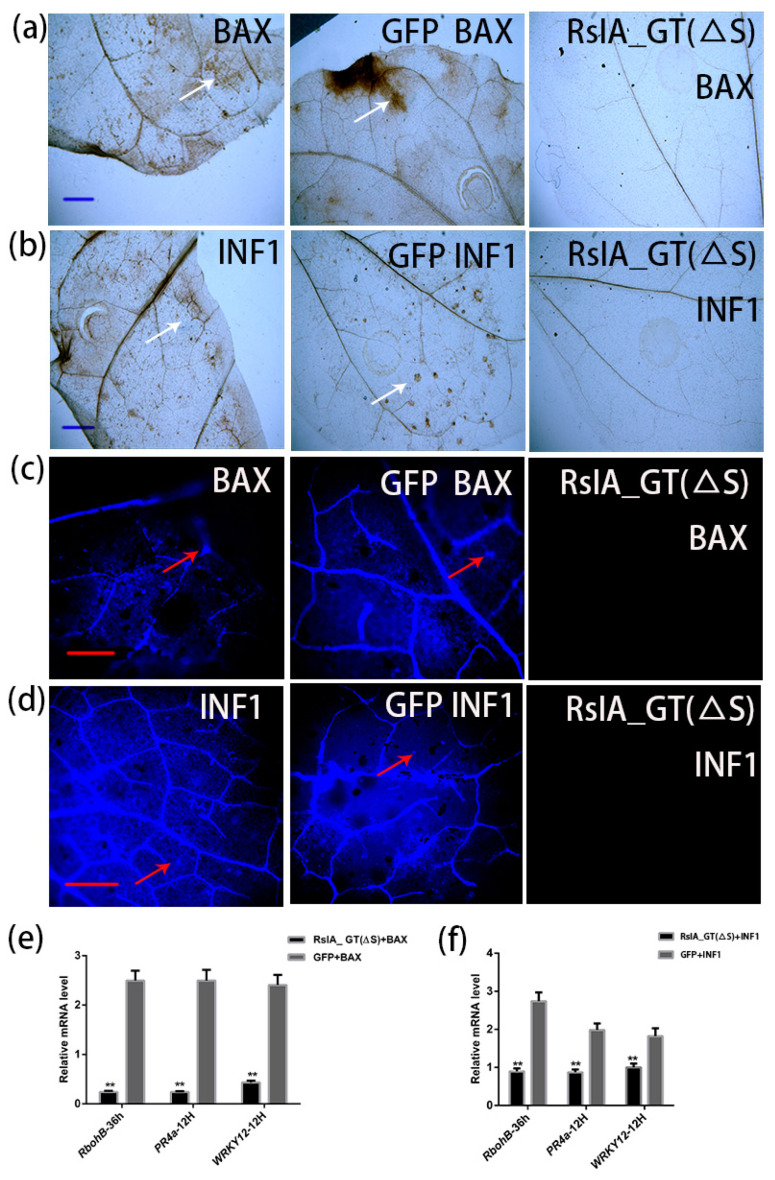
RsIA_GT(ΔS) suppresses the product of H_2_O_2_ and callose as well as the expression of the defense-related genes induced by *BAX* and *INF1* in *N. benthamiana*: (**a**–**d**) Detection of H_2_O_2_ by DAB staining method (**a**,**b**) and callose by aniline blue staining method (**c**,**d**) in the leaves agroinfiltrated with *BAX* (left), *BAX* and *GFP* (middle), and *BAX* and RsIA_GT(ΔS) (right). GFP served as a negative control. The leaves were sampled at 12 h (**a**,**b**) and 24 h (**c**,**d**) after the agroinfiltrations and used for the staining experiments, respectively. Bar (a and b), 2 mm. Bar (**c**,**d**), 10 μm. The white and red arrows indicate the representative areas for the accumulations of H_2_O_2_ and callose, respectively. (**e**,**f**) Expression analysis of defense-related genes, *RbohB*, *PR4a*, and *WRKY12* in *N. benthamiana* leaves agroinfiltrated with *BAX* and RsIA_GT(ΔS) (**e**) and *INF1* and RsIA_GT(ΔS) (**f**). *GFP* was used as a negative control. The leaves were collected at 12 h and 36 h after the agroinfiltrations and used for RNA extraction to synthesize cDNA for the qRT-PCR analysis. The experiments were repeated three times with independent biological samples. ** *p* < 0.01, Student’s *t*-test.

**Figure 6 pathogens-11-01026-f006:**
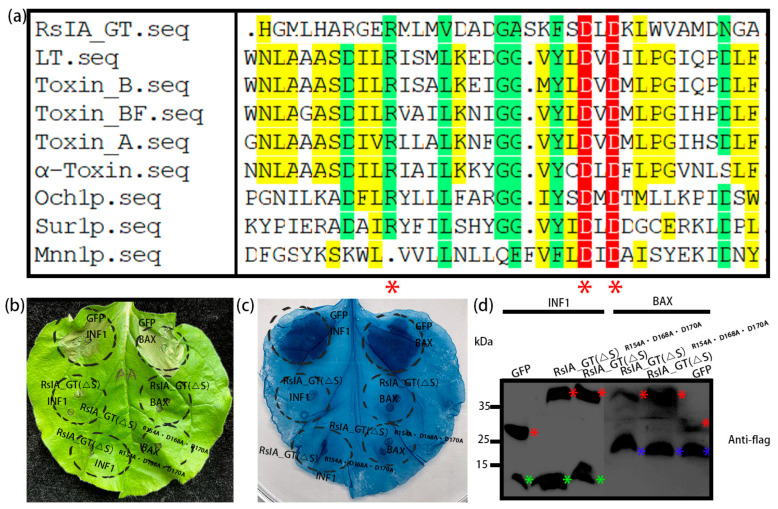
Conservation of key amino acid residues of glycosyltransferase catalytic domain in RsIA_GT and assessment of their requirements for the suppression activity of HR cell death: (**a**) Alignment of the amino acid sequences of glycosyltransferase catalytic domain of RsIA_GT and eight large clostridial toxins: LT (X82638 from *Clostridium sordellii* strain 6018), Toxin_B (X53138 from *Clostridioides difficile strain* VPI 10463), Toxin_BF (Z23277 from *C. difficile* strain 1470), Toxin_A (M30307 from *C. difficile strain* VPI 10463), α-Toxin (Z48636 from *Clostridium novyi*), Ochlp (D11095 from *Saccharomyces cerevisiae*), Surlp (M96648 from *S. cerevisiae*), and Mnnlp (L23753 from *Saccharomyces cerevisiae*). The red asterisks indicate the key amino acid residues of glycosyltransferase catalytic domain. (**b**,**c**) Co-expression analysis of RsIA_GT(ΔS), RsIA_GT(ΔS)^R154A D168A D170A^, and GFP with INF1 or BAX in *N. benthamiana* leaves for the evaluation of the requirement of the key amino acids for the immune suppression activity of RsIA_GT(ΔS). The indicated combinations of genes were agroinfiltrated, and necrosis (**b**) and cell death detected with trypan blue staining (**c**) were observed at three days after the treatment. GFP served as a negative control. (**d**) Western blot confirming the co-expression of RsIA_GT(ΔS), RsIA_GT(ΔS)^R154A D168A D170A^, and GFP with BAX or INF1 using an anti-FLAG antibody. The blue asterisks indicate the protein bands of BAX. The green asterisks indicate the protein bands of INF1. The red asterisks indicate the protein bands of the correct size.

**Figure 7 pathogens-11-01026-f007:**
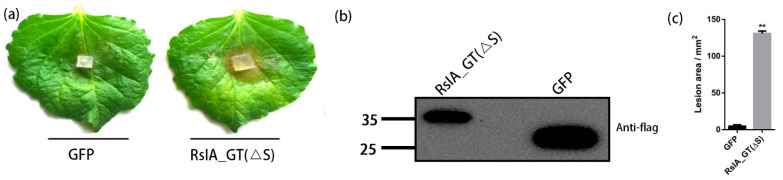
Transiently expressed RsIA_GT(ΔS) enhanced the symptom formation caused by *R. solani* AG-1 IA in *N. benthamiana*: (**a**,**b**) *R. solani* AG-1 IA was inoculated to the leaves of *N. benthamiana* at 2 days after the filtration with Agrobacterium harboring RsIA_GT(ΔS) or *GFP*. The lesion formation was observed at three days after the inoculation. (**c**) The lesion area was quantified using the ImageJ software. Data were mean ± SE of three independent biological replicates (** *p* < 0.01, Student’s *t*-test).

## Data Availability

Data are contained within the article or Appendix A.

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
