# Peer review of "Secreted Glycosyltransferase RsIA_GT of Rhizoctonia solani AG-1 IA Inhibits Defense Responses in Nicotiana benthamiana"

_pathogens, 2022, doi:10.3390/pathogens11091026_

Round 1

Reviewer 1 Report (Previous Reviewer 1)

All my suggestions have been well addressed.  

Author Response

Dear Reviewer,

Thank you for giving us the opportunity to submit a revised draft of the manuscript for publication in Pathogens. We appreciate all the time and effort that you dedicated to providing feedback on our manuscript and are grateful for the insightful comments on and valuable improvements to our manuscript.

According to the comments of Editors and other reviewers, we made minor revisions to the newly submitted manuscript. Those changes are highlighted within the manuscript. We hope you will find this revised version satisfactory.

Thanks again for your time and effort and look forward to hearing from you.

Sincerely, 

Danhua Zhang, PhD

Reviewer 2 Report (Previous Reviewer 2)

The authors included all the suggestions, and this version of the manuscript has improved considerably compared to the previous one. 

Author Response

Dear Reviewer,

Thank you very much for your time involved in reviewing the manuscript and your very encouraging comments on the merits. These suggestions have enabled us to improve our work.

According to the comments of Editors and other Reviewers, we made minor revisions to the newly submitted manuscript. The revised portions are marked in red in the manuscript. We hope you will find this revised version satisfactory.

Thank you again for all your time and this great opportunity for us to improve the manuscript. We look forward to hearing from you soon.

 Sincerely,

Danhua Zhang, PhD

Reviewer 3 Report (New Reviewer)

The authors the characherize a glucosyltranferase secreted by the fungus Rhizochonia solanis khin that plays a role in pathogenicity and its expression increases during early infection of the plant rice.

Thd introduction is relatively long considering other parts of the manuscriot are long. While it is important to show the diversity of the glyconoyltransfterase in other species it can be summarized, Adding so much detail from other species distracts attention from the current study.

The authors may consider sticking to the format prescribed in instruction to author ie introducuction, methods materiala, results discussion and although not mandatory a conclusion will be warranted in this particular paper.

Author Response

Dear Reviewer, 

Thanks for the time and effort that you have put into reviewing the previous version of the manuscript. We appreciate the detailed and constructive comments which have greatly helped us improve our manuscript. 

Based on the comments and suggestions provided, the English has been improved throughout the manuscript, the introduction has been shortened, and the conclusions have been added. We have uploaded the file of the revised manuscript with all the changes highlighted by using the track changes mode in MS Word. 

We would also like to thank you for allowing us to resubmit a revised manuscript. We hope this revised manuscript has addressed your concerns and look forward to hearing from you.

Sincerely, 

Danhua Zhang, PhD

This manuscript is a resubmission of an earlier submission. The following is a list of the peer review reports and author responses from that submission.

Round 1

Reviewer 1 Report

The manuscript entitled "A glycosyltransferase RsIA_GT of Rhizoctonia solani AG1-IA inhibits plant innate immune response" is a good piece of work which highlights the regulatory role of glycosyltransferase RsIA_GT from Rhizoctonia solani AG1-IA on host defense response, in terms of suppressing cell death, H2O2 production, callose accumulation, and genes related to defense responses in N. benthamiana. In my opinion the overall concept is interesting and important. I have a few suggestions which I believe will improve the manuscript.

Comment

  1. The abstract should illuminate the main findings of the paper that can serve as a stand-alone document. However, authors have represented abstract in more generalized form. Authors should emphasize the levels of decrement of different parameters assessed in % values.
  2. Line 76-77 The sentence “Among the identified potential pathogenesis proteins, which are small secreted proteins (<400 aa) and were upregulated at different inoculation times”is not clear, please rewrite.
  3. Line 135 “could to trigger”change to “could trigger”
  4. English should be improved; grammar needs enhancement in many sentences and paragraphs.

Author Response

Dear Reviewer, 

Thanks very much for taking your time to review this manuscript. I really appreciate your comments and suggestions! Your comments and suggestions have given me great confidence in my research. I am particularly grateful to you for pointing out my shortcomings in understanding and writing abstracts.

Please find my itemized responses in the attachment  and my revisions in the re-submitted files.

Sincerely,

Danhua

Reviewer 2 Report

In this work, entitle “A glycosyltransferase RsIA_GT of Rhizoctonia solani AG1-IA inhibits plant innate immune response”, the authors report the identification of a glycosyltransferase which display a signal peptide sequence that could suppress the host immunity promoting the virulence of plant pathogens.

In general, the manuscript is very interesting and is well documented.

Some comments and suggestions for the manuscript are listed below.

Introduction:

Line 52: “Recently, several fungal pathogens glycosylation pathways pathogenesis have been identified “. Include reference.

Why is important to evaluate the signal peptide? It is necessary to improve the introduction according to the main goal of the work which is to validate the importance of the signal peptide of RsAI_GT.

Materials and methods

Line 256 and 258: To correct “colii

To correct the supplementary material: “velidation”

Author Response

Dear Reviewer, 

Thanks very much for taking your time to review this manuscript, and your affirmation and praise of my manuscript. I really appreciate your comments and suggestions! Thank you very much for pointing out my serious omission of background introduction related to the experimental results.  

Please see my itemized responses in the attachment and my revisions in the re-submitted files.

Sincerely, 

Danhua

Reviewer 3 Report

The manuscript of Zhang et al., entitled “A glycosyltransferase RsIA_GT of Rhizoctonia solani AG1-IA 2 inhibits plant innate immune response” describes the “identification” of a transferase which is then tested for its potential function in rice, and N. benthamiana.  In M&M various items are mentioned not used (?) in the manuscript. Please provide evidence by mutating the catalytic domain of the protein and show that this is the cause of the induced necrosis/yellowing in rice or the effect on BAX and Inf induced cell-death in N. benthamiana.

There is also a statement that the studied protein had GT activity. Please show this activity or analyse the catalytic site in great detail in order to can say this.. This is speculation. If in Figure 1 the distances are so big, you have to validate this per protein.

The manuscript is extremely poor in writing. Nearly every sentence has issues either with writing itself or the lack of essential details which makes the manuscript unpublishable. Hereafter a large number of the comments and additional questions as examples of this, so by far not covering all issues encountered in the manuscript. I would recommend additional work together with rewriting by a native English writer with sufficient scientific background to be precise.

Abstract with issues discussed between <>.:

 Rhizoctonia solani AG1-IA <what is this, a strain, a pathotype, explain> has a wide range of hosts and seriously threatens crop yield<s>. Various transferases <all, glycotransferases, other?> secreted by fungus <fungi or a fungus>play an important role. While, there are no reports on the transferase of R.  solani AG1-IA, and their pathogenesis <of the transferase?> is poorly understood. In the predicted secreted proteins <predicted from what?>, we found a glycosyltransferase, RsIA_GT (AG11A_09161), highly conserved in R. solani <of course, you mean among R. solani strains or groups...> and could cause lesion and yellowing in rice <What causes the lesion and yellowing, how? expression? infiltration.. this is not clear enough even for an abstract>. Our study found that RsIA_GT possesses a signal peptide (SP) in the N-terminal <ever seen a signal peptide in another region of a protein> and could lead proteins out of cells <lead proteins out of cells is so incorrect language>. Both the full-length sequence RsIA_GT and the de-signal <de-signal? With and without signal peptide..> peptide sequence RsIA_GT(â–³S) could not induce cell death in Nicotiana benthamiana leaves. RsIA_GT(â–³S) could suppress the  cell death induced by BAX and INF1 in N. benthamiana. The H2O2 production, callose accumulation, and  genes related to defense responses induced by BAX and INF1 were also suppressed by RsIA_GT(â–³S) <nice but Bax and Inf are not from R. solani, so what does it mean?>. Transient expression of RsIA_GT(â–³S) <you transiently express a gene, so in Italics> would make N. benthamiana more susceptible to R. solani AG1-IA. Overall, our results reveal a glycosyltransferase, RsIA_GT, which could suppress hosts' innate immunity and promote virulence of pathogens.

Other points grammatically and scientifically (note, by far not all encountered...)

In the introduction again many grammatical mistakes: Line 29: solani is A basidiomycete, not the
same line: pathogen that infects, not pathogen infects..

Line 31, how can you infer that different anastomosis groups are related to a high genetic diversity?.

Line 35, Is this the moment of infection as suggested or when it becomes visible. (tillering/stem elongation)

Why introduce in line 42 Golgi apparatus as AG instead of the well used GA..

Lines 49/50, words as kinds and different should be avoided.

Line 52: Recently... the manuscripts referred to are from 2012 (!) to 2019, that’s hardly recently.

Line 75-78, this should be introduced in the introduction, not in the results.. what is the relation between predicted potential secreted proteins, and the potential pathogenesis proteins? If you give a number for one of them, you have to indicate it also for the other, how many are <400 AA? Etc..

How is Figure 1 produced, AA alignment or NT alignment. Only talking on identity, not on similarity....? How that was performed is not present in the methods section..

Figure 1 is interesting, are these the closest homologs in the database of were many skipped?

Line 90 “ out of the cell” is not scientific...

Figure 2a: Using Signal-P (version 3.0) with this sequence the prediction for a signal anchor instead of signal peptide is made.. There is a lack of confidence in Signal-P version 5.0 and there is already Signal-P version 6.0...which does not judge it as SP... (why not update your figure with the latest version) so how can you prove this (his tagged pull down of extracellular medium?)... Signal anchors also generate such result in the yeast system. This would be a reason that the color reaction staining is reduced.

General comment, please check throughout what you mean with e.g. the sp was fused... You cannot fuse a SP but you can fuse the sequence encoding the SP to another sequence and produce a chimeric protein..

Line 102: Avr1b served as the positive control. Pretty sure you only used the SP in this case.... so be precise!. (line 109, the N-terminalS of both genes...)

Line 118, the difference in color in the reaction might be due to the fact that invertase is not released in the medium but sticks with a signal anchor (see comment above) and then you get less color... Note that also in N. benthamiana there is less response... same reason...

Line 130: what is the coding sequence? Does this include or not include the signal peptide or lacks it other parts...? Be precise...

Line 134, a protein is not virulence... incorrect grammatics...

Line 134: The control was also prokaryotic expressed products?? Which?? Minimum would be MBP as control, or RsIA_Gt with signal peptide.  Most properly, the catalytic domain is mutated, so this activity dead mutant would not induce the effects...

Line 143: Crude protein? what is crude protein? is this the MBP protein, a Flag tagged (mentioned in M&M but not in the main text), or extracellular medium from the pathogen?

Talking about the FLAG-tag, Where is the flag-tag western blot...? which would make life so much easier.

Line 147: The protein is not virulence.... it is necrosis and yellowing inducing..
Line 147: here and on other locations, you transiently express a gene resulting in transient production of the protein.

Figure 4: Please provide also the trypan blue staining for the C figures

Line 167; Knowing that the SP is necessary to be secreted by the pathogen, what would be the reason for a protein without SP to inhibit BAX and INF1? Issue with location, uptake etc....

Line 174: Since Nicotiana benthamiana is not tobacco (Nicotiana tabacum), you cannot use tobacco as a general term if you did not use tabacum.

Line 203: “RsIA_GT could promote the virulence” This is NOT true, thus far only  the protein product without signal peptide can do it. It is positioned at a wrong location, potentially with wrong protein modifications and inhibits somehow (e.g. blocking the system) the induced responses.. Without additional evidence this is way to far to state this comment (on top of this, that's a conclusion, not a result)!

Line 215-216” Our study find a secreted protein, RsIA_GT(â–³S), having glycosyltransferases activity”, There is no evidence in the whole manuscript that the proteins is shown this activity. This is pure speculation.

Line 240-242: “ In our study, the transient expression of RsIA_GT(â–³S) in N. benthamiana enhancing the virulence of R. solani AG1-IA may be  caused by the N-glycosylation of effectors of the pathogen.”

Line 268-269, Maybe overlooked but where was a GFP version used? Also no picture in the manuscript on subcellular localization.. Is there any indication that after secretion of these proteins, they are taken up by the plant cell and await effectors from the pathogen to be N-glycosylated... very far fetched.. If so, the variant with signal peptide should work too...

Similar, in the M&M the MBP construct is not mentioned but instead, FLAG is used...? This does not match with the results section.

Author Response

Dear Reviewer,

Thanks very much for taking your time to review this manuscript. We really appreciate your comments and suggestions!

According to the comments and suggestions, we added a lot of background introductions related to this study. In addition, your rigorous and sincere comments greatly improved the rigor and accuracy of the words we used to describe scientific issues. It also greatly facilitated our comprehensive, accurate and in-depth analysis of experimental results. We appreciated for your warm work earnestly!

Please see my itemized responses in  the attachment and my revisions in the re-submitted files.

Sincerely,

Danhua

Round 2

Reviewer 3 Report

In my first review, I indicated that my comments about English writing were the tip of the iceberg. Indeed, in the resubmitted manuscript, my stated points have been corrected, but the rest is still embarrassing. So the general data remained largely the same and might have improved a bit, but the manuscript as it is cannot be judged by its general value. Unless an actual corrected version is submitted, I will not spend furhter time to generate a full review that consider's the (final) presented data rather then being a proofreader for textual issues.